

# Carbon Storage in Coastal Reed Ecosystems

Margaret F. Williamson[1], Tom Jilbert[2], Alf Norkko[1], Camilla Gustafsson[1]

[1]Tvärminne Zoological Station, University of Helsinki, Hanko, 10900, Finland
[2]Department of Geosciences and Geography, University of Helsinki, 00014, Finland

*Correspondence to*: Margaret F. Williamson (margaret.williamson@helsinki.fi)

**Abstract.** Common reed (*Phragmites australis*) distribution has increased in coastal ecosystems across the globe, including around the Baltic Sea. Understanding carbon (C) storage in reed beds is critical for developing more accurate blue carbon (BC) budgets and best management practices, yet there currently appears to be a gap in knowledge about C cycling in these ecosystems. Reed beds are typically categorized as salt marsh ecosystems in BC budgets, but preliminary findings indicate

reed beds are unique from salt marsh ecosystems and show great potential for C storage. It is, therefore, important to understand C storage in reed beds so that these ecosystems can be taken into consideration while developing BC budgets.

The aim of this spatial study was to quantify how much C is stored in above- and belowground biomass, and sediments in the different zones of reed beds along the Pojo Bay system of the northern Baltic Sea in coastal Finland. We selected 6 reed bed sites to sample along Pojo Bay from the northern-most part of the Bay to the southern-most part opening into the Baltic

Sea, covering a range of salinities and wave exposure. In each site, samples were selected randomly within each of the 3 reed bed zones (terrestrial, intermittent, and littoral) and analyzed for sediment parameters (dry bulk density, organic matter content), plant characteristics (stem density), and plant and sediment-bound C content.

Sediment samples were collected down to 1m depth, when possible, with the use of Russian peat borers and a box corer. Dry bulk density (DBD) was variable across all sites, sediment depths, and reed bed zones with highest DBDs measuring 2.55 g

cm$^{-3}$ (LOI (loss on ignition) = 0.7% and water = 1.3%) and the lowest DBD measuring 0.05 g cm$^{-3}$ (LOI = 74% and water = 94.6%). The results from sediment LOI show higher organic matter content in the upper 30 cm of sediment profiles and a general trend towards higher organic matter content in terrestrial and intermittent zones than littoral zones of reed beds. C content in sediment, above- and belowground biomass was significantly different at the zone level for all sites and, with one exception, was significantly different at the site level for all variables measured (DBD, LOI, sediment C stocks, stem counts,

aboveground C stocks, and belowground C stocks). The highest sediment C stocks were typically found in the intermittent zone while the lowest were typically found in the littoral zone across all sites. Average stem counts were variable across sites and reed bed zones with the highest and lowest stem counts being 217 stems m$^{-2}$ and 9 stems m$^{-2}$, respectively. Aboveground biomass C stock averages were generally highest in the intermittent zones and lowest in the terrestrial and littoral zones. Belowground biomass C stock averages were generally highest in the intermittent zone and lowest in the

littoral zone. C storage in reed bed sediments and belowground biomass was higher than C storage in aboveground biomass for all sites.



These findings are significant as they help rectify a gap in knowledge on how much C is stored in reed bed biomass and sediment which is important for management of this rapidly expanding coastal ecosystem type and enables researchers to develop more accurate coastal carbon budgets to combat climate change.

## 1 Introduction

Vegetated coastal ecosystems are important contributors to global C sequestration (Buczko et al. 2022, Macreadie et al. 2019, Duarte et al. 2013, Jingtai et al. 2022) and coastal ecosystems that capture and store BC include seagrass meadows, tidal marshes, and mangrove forests (Macreadie et al. 2019, IPCC 2013, Howard et al. 2014). These ecosystems are a critical component of the global C budget and are essential in plans to combat climate change (Gilby et al. 2020, Kang et al. 2023, Macreadie et al. 2021, Mazarrasa et al. 2023). The common reed (*Phragmites australis*) is a wetland grass species that grows rapidly and is able to outcompete other vegetation in wetlands and coastal ecosystems throughout the world (Newton et al. 2020, Adams 1999). It is native to coastal areas of the Baltic Sea though changes in management practices have led to increased distribution of reeds in areas such as the Baltic Sea coast (Niemi et al. 2023) and beyond (Meriste et al. 2012, Huhta 2009).

Despite the expansion of the common reed in coastal ecosystems over recent decades, previous research on the role of reed beds in C cycling is fairly limited (Moore et al. 2012, Caplan et al. 2015, Duke et al. 2015, Mozdzer & Zieman 2010). Research to date indicates that C stocks in reed beds could average around 17.4 kg C m$^{-2}$ in comparison to the estimated global average for C stocks in tidal salt marshes of approximately 25 kg C m$^{-2}$ (Pendleton et al. 2012), demonstrating that reed beds play an important role in C storage and that further research into their storage capacity is needed to improve the precision of the coastal C budgets (Buczko et al. 2022, Silan et al. 2024).

Increased reed presence in coastal environments has also led to more interest in reed management practices, including ones financed by the European Union such as BalticReed. If reed beds are cut back in a way that disturbs root systems and discharges large amounts of sediment into water bodies, the C stored in these ecosystems could be released (Huhta 2007, Güsewell 2003). Additionally, careful management of reed beds could lead to improved C storage capacity (Cui et al. 2022), thus C storage in reed beds is important to take into account when developing best management practices for reed beds in order to avoid excessive C release from these ecosystems. Careful reed management is of particular importance to coasts of the Baltic Sea where eutrophication poses a large challenge as release of sediments into the water can increase nutrient discharge into the sea (Pawlak et al. 2009, Furman et al. 2014).

Increased interest in reed bed management means there is a greater need to understand the potential role these ecosystems play in C cycling. Despite the prevalence of reed beds, there is insufficient knowledge on how much C is stored in these systems and how it changes along environmental gradients such as varying surface water salinities, wave exposures, and topographical zones of the reed beds (i.e., the terrestrial zone, intermittent zone, and littoral zone). Current evidence suggests that these factors impact reed growth, filtration rates in reed beds, and C storage efficiency (Altartouri et al. 2014, Asaeda et



al. 2003, Silan et al. 2024). Hence, the aim in this study was to investigate C storage in different parts of the reed bed
ecosystem across different environmental gradients such as salinity and wave exposure. We hypothesized that reed bed C
storage would differ between sites and reed bed zones and that the highest rates of C storage would be found in reed bed
sediments.

## 2 Material and Methods

### 2.1 Study Area

We conducted our study in reed beds in the well-described archipelago system of the microtidal Baltic Sea along gradients in
the Pojo Bay system, Finland. This area has been noted as one of the largest reed-growing areas in Finland (Pitkänen 2006).
Six sites were chosen within Pojo Bay covering a range of salinities and wave exposure from the northern-most part of the
Bay to the southern-most part opening into the Baltic Sea (van der Meijs & Isaeus 2020, Iaeus & Rygg 2005, Figure 1). The
six sites are spaced approximately 6-8 km apart, are distributed along both sides of Pojo Bay, and range in size from an area
of approximately 0.01 km$^2$ to 0.3 km$^2$. All sites were accessible by boat and by car (Table S1).

### 2.2 Experimental Design and Field Sampling

C stocks in *Phragmites* were assessed from sediment and biomass samples taken in August 2023. Reed bed perimeters and
their separate zones were delineated at each site prior to sample collection campaigns with the use of the ArcGIS Field Maps
App, ArcGIS Online, and ArcGIS Pro. Zones within the reed beds were determined by the following parameters: (1) the
terrestrial zone indicators included dry land, grass, trees, and/or rocks; (2) the intermittent zone indicators included saturated
soils, lack of aquatic vegetation, and/or presence of algal species on top of rocks; and (3) the littoral zone indicators included
standing water, presence of aquatic vegetation, and/or open water. Additionally, topographic changes in each reed bed
usually coincided with changes in zone type.

Wave exposure gradients were estimated at the site level using data layers from the Finnish Environment Institute (SYKE)
open source library which used a nested-grid that includes finer grids with input fetch values to model wave exposure for the
Finnish coast (Iaeus & Rygg 2005). Smaller-scale wave exposure gradients are also present within each of the sites between
the three reed bed zones as the littoral zones are along the water's edge of the reed beds and are the most exposed zone, the
intermittent zone is further inland and more sheltered, and the terrestrial zone is the most sheltered as it is along the land's
edge of the reed beds (Buczko et al. 2022). See Table S1 for details about wave exposure for each of the zones at each site.
A random starting point for sampling was selected in each of the three zones of each site by placing a 1m grid overlay on
maps of the reed beds created in ArcGIS Pro and using multi-sided dice to determine grid numbers (eg. grid number 012
across and 006 up). Replicate locations were then selected along a transect within each zone at 3-5 m intervals from the
starting point, resulting in 3 replicates per zone. 1m x 1m quadrats were used at each of the 9 sampling points per site and all
*Phragmites* stem counts, sediment samples, and biomass samples collected during field campaigns were chosen from within



the 1m x 1m quadrats. A tape measure was used for stem measurements and depth of standing water within the quadrat. Datasheets for stem counts were generated in Survey123 using ArcGIS Online.

Sediment samples were collected in each of the 9 sample quadrats using either a Russian Peat Borer to 50 cm depth or a box corer to 100 cm depth (Howard et al. 2014, Serrano et al. 2019). Soil cores were divided into subsamples of 10 cm depth intervals. Different coring devices were used in each of the three reed bed site zones. The box corer was used for sampling in

the terrestrial and intermittent zones whenever possible, while the Russian peat borer was used for sampling in the littoral zones where longer handles were required to reach sediments through deep standing water. The Russian peat borer was also used in terrestrial and intermittent zones where the box corer did not work due to rocky terrain, sandy soils, or other challenging conditions. Sediment core lengths differed between sampling points based on which soil corer could be used and how far down the soil corer could go until either bedrock was reached or sand content or clay layers made deeper penetration

by hand powered corers impossible. Sediment C stocks were integrated across the whole core and the organic material rich layer (when LOI was less than 5%).

Biomass samples of the reed were collected in the field using quadrats. Within each 1m x 1m quadrat, 12 healthy plants were selected from the corner of the quadrat closest to the person collecting the samples (Xiong et al. 2022, Howard et al. 2014). If less than 12 healthy plants were present within the quadrat, then all plants were collected. Belowground biomass samples

were collected using soil corers. The soil core samples collected for belowground biomass were sieved in the field using 1 mm sieve trays, when possible (Dong et al. 2012, Howard et al. 2014).

**2.3 Aboveground and Belowground Biomass Processing**

Soil cores collected for belowground biomass that contained large amounts of clay required soaking in water for 24 h back in the laboratory and then additional sieving using 1 mm sieve trays (Hillmann et al. 2020). All biomass samples were rinsed

clean back at the laboratory, weighed, and dried at 70°C for 72 h until they reached a constant weight. Samples were then ground up and C content was determined using a Thermo Finnigan DELTA$_{plus}$Advantage continuous-flow stable isotope-ratio mass spectrometer (CF-SIRMS) coupled with a FlashEA 1112 elemental analyzer (Thermo Electron, Bremen, Germany) at the University of Jyväskylä in Finland. Standards used for analysis of above- and belowground samples included birch leaf lab standards, USGS88, and USGS91.

**2.4 Sediment Analysis**

Each sediment subsample was bagged in the field, brought back to the laboratory and weighed fresh, then dried at 70°C for 72 h until they reached a constant weight. Samples were then ground up and measured for organic matter content using the LOI method where samples were combusted in a muffle furnace at 450°C for 4 h. DBD of sediment samples was calculated based on LOI-based corrections for sediment density: sediment density = (1.4 * (LOI/100) + 2.65 * (1-(LOI/100))). In this

method, wet weights and dry weights were used to calculate sediment content which was then converted to sediment volume and porosity. Wet bulk density (WBD) and DBD were then calculated using porosity, sediment density, and sediment





content. C analysis (total organic carbon, TOC) was also run on 127 of the total 305 sediment samples and analyzed at the University of Jyväskylä in the same manner as mentioned above for biomass. Sediment C stocks were calculated using TOC, LOI, and DBD. Sediment C stocks were estimated using LOI data for the sediment samples where TOC was not available.

## 2.5 Statistical Analysis

All statistical analyses were done using R Statistical Software (v4.3.2, R Core Team 2024). A variety of data exploration devices were implemented (Zuur et al. 2010). Due to the small sample size (54 observations for data related to each of the variables being measured), a Shapiro-Wilk's test and histograms were used to test for normal distribution of data results (DBD, stem counts, LOI results, sediment C stocks, and above- and belowground biomass C stocks). All data sets were found to be different from normal distribution and an additional Fligner-Killeen test was used to test for homogeneity of the non-normal data. Fligner-Killeen results showed all data were significantly different from homogeneity meaning the variance of the residuals was unequal (heteroskedasticity). Due to the heteroskedastic and non-normal nature of the data, analysis was conducted using PERMANOVA via the vegan package (Oksanen et al. 2020). PERMANOVA tests also allowed us to deal with the nested nature of the zones within each site and were used to determine significant differences in data at the site and zone levels. PERMANOVA results with p values < 0.05 were found to be statistically significant. PERMANOVA post hoc tests were used to further determine how C stocks differed between zones across sites. All averages are presented with standard error.

## 3 Results

### 3.1 Environmental and Sediment Variables

Surface water salinity during sampling ranged between 0.68 ppt and 6.15 ppt at the different sites (Figure 1). Sampling sites consisted of dense stands of *Phragmites australis* with a few individual stems of *Typha angustifolia* and *Schoenoplectus lacustris* found in the littoral zone of site 1.

The average soil core length across all zones was 56.5 ± 3.2 cm while the average for each zone was 67.2 ± 6.4 cm (terrestrial), 58.3 ± 5.8 cm (intermittent), and 43.9 ± 2.2 cm (littoral). We detected statistically significant differences at the site and zone level for DBD (Tables 1 and 2). DBD was variable across all sites, sediment depths, and reed bed zones (Figure 2). The highest DBD was 2.55 g cm$^{-3}$ (LOI = 0.71% and water = 1.3%) and lowest DBD was 0.05 g cm$^{-3}$ (LOI = 74% and water = 94.6%). The average bulk densities per site and zone ranged from 0.64 ± 0.04 g cm$^{-3}$ to 1.60 ± 0.12 g cm$^{-3}$ (Table S2).

We detected statistically significant differences at the site and zone level for LOI (Tables 1 and 2). The highest sediment organic matter percentages (LOI results, shown here as % OM) were found in the upper 30 cm of soil profiles and the lowest overall organic matter percentages were typically found in the sediments of littoral zones (Figure 3). The average LOI results per site and zone for the full depth of the soil cores ranged from 0.5 ± 0.1% OM in the littoral zone for site 5 to 40.7 ± 8.9%



OM in the intermittent zone for site 6. The average LOI results per site and zone for the upper 10cm of the soil cores ranged from $0.7 \pm 0.2\%$ OM in the littoral zone for site 5 to $74.2 \pm 0.5\%$ OM in the intermittent zone for site 4 (Table S3).

**3.2 Sediment C Stocks**

We detected statistically significant differences at the site and zone level for sediment C stocks (Tables 1 and 2). Sediment C stocks were higher than belowground and aboveground biomass C stocks at all sites (Figure 4). The highest sediment C stocks were typically found in the intermittent zone while the lowest were typically found in the littoral zone (Figure 4). The average sediment C stocks per site and zone ranged from $1,566.7 \pm 409.2$ C g m$^{-2}$ in the littoral for site 5 to $24,408.3 \pm 1,607.4$ C g m$^{-2}$ in the intermittent zone for site 1 (Table S4). In our post hoc PERMANOVA tests, we found that sediment C stocks were significantly different between terrestrial and intermittent zones, between terrestrial and littoral zones, and between intermittent and littoral zones.

**3.3 Plant Variables**

We detected statistically significant differences at the site and zone level for stem counts (Tables 1 and 2). Stem counts tended to be higher in the intermittent zone at sites 1, 2, and 6 and in the littoral zone at sites 3, 4, and 5 (Figure 5). Highest stem counts for all sites ranged from 71 to 217 stems m$^{-2}$ with the highest stem count recorded in the littoral zone at site 5.

We detected statistically significant differences at the site and zone level for aboveground C stocks (Tables 1 and 2). Plant C stocks were higher in belowground biomass than aboveground biomass at all sites (Figures 6 and 7). Aboveground biomass C stocks were highest in the intermittent zones for sites 1, 2, and 6, highest in the littoral zones for sites 3 and 5, and were similarly high in the terrestrial and littoral zones for site 4. The aboveground biomass C stocks were lowest in the intermittent zone for site 4, lowest in the terrestrial zone for sites 3 and 5, lowest in the littoral zone for sites 1 and 6, and were similarly low in the terrestrial and littoral zones for site 2 (Figure 6). Aboveground biomass C stock averages per site and zone ranged from $24.6 \pm 3.3$ C g m$^{-2}$ in the intermittent zone at site 4 to $244.9 \pm 37.2$ C g m$^{-2}$ in the intermittent zone at site 6 (Table S5). In our post hoc PERMANOVA tests, we found that aboveground biomass C stocks were significantly different between the terrestrial and intermittent zones and between intermittent and littoral zones, but they were not significantly different between the terrestrial and littoral zones.

We detected statistically significant differences at the site and zone level for all belowground C stocks except for the belowground C stocks from the intermittent zones (Tables 1 and 2). Belowground biomass C stocks were highest in the intermittent zones of all sites except for site 6 where they were highest in the terrestrial zone. Belowground biomass C stocks were lowest in the littoral zone for site 1, 4, 5, and 6, lowest in the terrestrial zone for site 2 where belowground biomass samples were not obtained from the littoral zone, and were similarly low in the terrestrial and littoral zones for site 3 (Figure 7). Belowground biomass C stock averages per site and zone ranged from $75.0 \pm 37.6$ C g m$^{-2}$ in the littoral zone for site 5 to $6,883.6 \pm 769.2$ C g m$^{-2}$ in the intermittent zone for site 1 (Table S6). We had trouble retrieving belowground biomass



samples from the terrestrial and littoral zones at site 2 and were only able to retrieve one sample between these zones from
the terrestrial zone (2,289.41 C g m$^{-2}$).

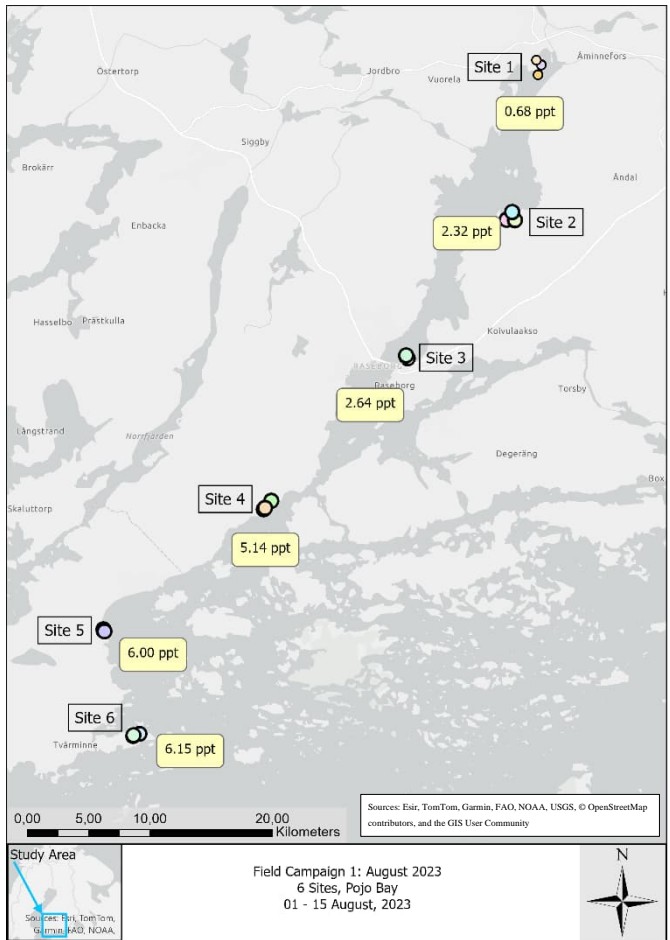

**Figure 1: Map of all reed bed sites and their surface salinity readings during sampling in August 2023.**






**Figure 2: Dry bulk density (g cm³) of sediment samples at varying depths across all reed bed zones and sites.**





**Figure 3: Loss on Ignition (LOI) results showing Organic Matter (%) in sediment samples at varying depths across all reed bed sites.**





**Figure 4: Carbon stocks (g m⁻²) in sediment all reed bed zones and sites. The box plot whiskers show minimum and maximum values, the horizonal line splitting the box shows the median value, and the box covers the interval where 50% of the data is found.**






**Figure 5: Stem counts per square meter across all reed bed zones and sites.**





**Figure 6: Carbon stocks (g m⁻²) in aboveground biomass across all reed bed zones and sites.**





**Figure 7: Carbon stocks (g m⁻²) in belowground biomass across all reed bed zones and sites.**



| Table 1. Differences at Site Level | Df | Sum Sq | F | Pr(>F) |
|---|---|---|---|---|
| **Dry Bulk Density** | | | | |
| terrestrial samples | 5 | 3.830 | 9.973 | 0.001 |
| residual | 115 | 8.833 | - | - |
| intermittent samples | 5 | 2.304 | 3.393 | 0.003 |
| residual | 99 | 13.447 | - | - |
| littoral samples | 5 | 1.080 | 5.448 | 0.001 |
| residual | 73 | 3.974 | - | - |
| **LOI** | | | | |
| terrestrial samples | 5 | 5.625 | 7.207 | 0.001 |
| residual | 115 | 17.952 | - | - |
| intermittent samples | 5 | 7.021 | 7.788 | 0.001 |
| residual | 99 | 17.852 | - | - |
| littoral samples | 5 | 5.898 | 15.508 | 0.001 |
| residual | 73 | 5.553 | - | - |
| **Sediment C Stocks** | | | | |
| terrestrial samples | 5 | 0.761 | 9.159 | 0.001 |
| residual | 12 | 0.199 | - | - |
| intermittent samples | 5 | 1.115 | 7.514 | 0.001 |
| residual | 12 | 0.356 | - | - |
| littoral samples | 5 | 0.727 | 6.285 | 0.001 |
| residual | 12 | 0.278 | - | - |
| **Stem Counts** | | | | |
| terrestrial samples | 5 | 0.899 | 12.212 | 0.001 |
| residual | 12 | 0.177 | - | - |
| intermittent samples | 5 | 0.756 | 7.581 | 0.002 |
| residual | 12 | 0.239 | - | - |
| littoral samples | 5 | 1.323 | 7.305 | 0.001 |
| residual | 12 | 0.435 | - | - |





| Table 1 (Continued). Differences at Site Level | Df | Sum Sq | F | Pr(>F) |
|---|---|---|---|---|
| Aboveground Biomass C Stocks | | | | |
| terrestrial samples | 5 | 0.923 | 4.125 | 0.012 |
| residual | 12 | 0.537 | - | - |
| intermittent samples | 5 | 1.740 | 26.904 | 0.001 |
| residual | 12 | 0.155 | - | - |
| littoral samples | 5 | 0.927 | 2.669 | 0.031 |
| residual | 12 | 0.834 | - | - |
| Belowground Biomass C Stocks | | | | |
| terrestrial samples | 5 | 2.293 | 7.028 | 0.002 |
| residual | 10 | 0.652 | - | - |
| intermittent samples | 5 | 0.688 | 1.642 | 0.122 |
| residual | 12 | 1.006 | - | - |
| littoral samples | 4 | 0.914 | 3.820 | 0.016 |
| residual | 10 | 0.598 | - | - |


**Table 1: PERMANOVA results showing how response variables varied across sites for all terrestrial, intermittent, and littoral zone samples (p < 0.05 statistically significant values indicated in bold).**





| Table 2. Differences at Zone Level | Df | Sum Sq | F | Pr(>F) |
|---|---|---|---|---|
| Dry Bulk Density | 17 | 11.924 | 7.997 | 0.001 |
| residual | 287 | 25.174 | - | - |
| LOI | 17 | 26.398 | 10.777 | 0.001 |
| residual | 287 | 41.356 | - | - |
| Sediment C Stocks | 17 | 4.217 | 10.717 | 0.001 |
| residual | 36 | 0.833 | - | - |
| Stem Counts | 17 | 3.683 | 9.168 | 0.001 |
| residual | 36 | 0.851 | - | - |
| Aboveground Biomass C Stocks | 17 | 4.023 | 5.583 | 0.001 |
| residual | 36 | 1.526 | - | - |
| Belowground Biomass C Stocks | 16 | 8.987 | 7.965 | 0.001 |
| residual | 32 | 2.257 | - | - |

**Table 2: PERMANOVA results showing statistically significant (p < 0.05) differences in response variables at the zone level accounting for the effect that site has on the zone**






## 4 Discussion

Understanding C storage in reed beds as they continue to increase their distribution across coastal ecosystems is imperative
for developing more accurate coastal BC budgets to combat climate change. The results show that these systems are unique
from the tidal salt marsh ecosystems they are generally categorized as in BC budgets. The results also highlight that C
storage is occurring in these systems at a rate that must be accounted for in reed bed management plans.

### 4.1 Dry Bulk Density and LOI

The differences we found in DBD and LOI results between zones and sites (Figures 2 and 3, Tables 1 and 2) are
understandable given the different environmental factors present at these levels. Differences in standing water depths
between the zones, stem counts between the zones and sites (and therefore the amount of rooting activity occurring in the
sediments), salinity levels between the sites, and anthropogenic activities between the sites and zones have each been shown
to impact soil physical properties and infiltration rates (USDA-NRCS 2019, Warrence et al. 2003). The soil cores obtained
during sampling showed organic rich top layers with either clays or sandy soils below, which also explains the inverse
relationships seen between the DBD and LOI results (Figures 2 and 3).

### 4.2 C Stocks in Coastal Reed Sediments

The differences in sediment C stocks between different zones and sites could be related to the differences in DBD and stem
counts we found at these levels. DBD, specifically, affects infiltration and organic matter and nutrient contents (USDA-
NRCS 2019) while stem counts impact the amount of *Phragmites australis* available for sequestering C in the reed bed
sediments through biogeochemical processes, reduced erosion, and increased filtration (Silan et al. 2024). Additional factors
that could be influencing sediment C stocks include anthropogenic activities, wave exposure, changes in soil characteristics,
to name a few.

Our findings are consistent with Howard et al. 2014 which theorized that most C storage in tidal salt marshes occurs in
sediments and belowground living biomass (roots and rhizomes). These findings are understandable as belowground C stock
systems are less prone to disturbance than aboveground systems (Fiala 1976). Our sediment C stocks averaged 7.9 C kg m$^{-2}$
in comparison with Buczko et al. 2022's rates of 17.4 C kg m$^{-2}$. Estimated global averages for C stocks in tidal salt marshes
are approximately 25 kg C m$^{-2}$ (Pendleton 2012), demonstrating that reed bed C storage is unique from the tidal salt marsh
ecosystems they are typically categorized as in current BC budgets and that further research into their storage capacity is
needed to improve the precision of the coastal C budgets (Buczko et al. 2022). The differences between the C stock averages
in our results and those of Buczko et al. 2022 could be related to differences between their sample sites in sheltered, brackish
lagoon systems along Germany's Baltic Sea coast and our sites in Finland which included sheltered and exposed sites and



occurred along a smaller salinity gradient. The German sites had tides and a salinity range from 0-10 PSU (ppt) while the Finnish sites had no tides and a salinity range from 0 to 6 ppt. It is not clear what time of year samples were collected in Buczko et al. 2022 but it could have differed from the August covered in our study. In addition, Buczko et al. 2022 notes that more research from other parts of the Baltic Sea are needed to better understand reed's role in BC storage in the Baltic Sea and the differences between their average sediment C stocks and ours further support this point.

### 4.3 Stem Counts and C Stocks in Coastal Reed Aboveground Biomass

The differences we found in stem counts between different zones and sites (Figure 5 and Tables 1 and 2) are understandable as differences in water depth, salinity, and presence of competing species (such as plants in the tree lines of the terrestrial zone) are known to impact reed growth (Altartouri et al. 2014, Asaeda et al. 2003). The differences we found in aboveground biomass C stocks between different zones (Figure 6 and Tables 1 and 2) could be due to the fluctuating nature of the terrestrial and littoral zones of reed beds. These zones represent the fringes of reed bed growth as reeds either start working their way into the open waters of the littoral zone to expand their territory or start competing with terrestrial plants and drier soils of the terrestrial zone (Pitkänen et al. 2007). The intermittent zones have the most sheltered environments for reed growth so it is understandable that aboveground biomass C stocks found there would be different from the terrestrial and littoral zone ones, but that terrestrial and littoral zones C stocks might not be different from each other. As discussed with the sediment C stocks, environmental factors that could be impacting aboveground biomass C stocks may include differences in anthropogenic activities, wave exposure, changes in soil characteristics, and others.

There has been an increased interest in reed bed management practices focused on removing aboveground biomass as a way of reducing nutrients in a system (e.g., Finnish-Swedish Interreg BalticReed project co-funded by the European Union), however previous research indicates aboveground biomass contains relatively low C content (Dong et al. 2012) and our findings support this. The density of aboveground biomass in reed beds can substantially impact erosion and water quality (Horppila et al. 2013) and management practices focused on removing aboveground biomass, such as mowing, can release nutrients stored in reed beds into the surrounding environment (Güsewell 2003) or even impact greenhouse gas fluxes (Rietl et al. 2017) so C storage in reed beds must be taken into account when developing management practices in coastal reed bed ecosystems.

### 4.4 C Stocks in Coastal Reed Belowground Biomass

The differences we found in belowground biomass C stocks between zones (Figure 7 and Tables 1 and 2) indicate that zonation is important to consider when accounting for C stocks in belowground biomass. This could be related to the buffered state of the belowground intermittent zones as these zones are better protected from processes such as wave exposure and anthropogenic impacts occurring aboveground and along the edges of the reed beds.

The differences in above- and belowground biomass-bound C stocks between zones is also interesting to note. Belowground biomass C stocks were typically highest in the intermittent zone while there was more fluctuation in zones with the highest



aboveground biomass C stocks. This finding could be related to the nature of the rhizosphere in reed beds. The terrestrial and
littoral zones of reed beds are the fringe zones where reeds are just starting to push into new territories. Reeds are capable of
producing stems up to 10m away from rhizomes located in a suitably moist habitat, enabling them to push into wetter and
drier areas than is preferable (Huhta 2007) so it is understandable that a more established root system in the intermittent zone
would lead to highest belowground biomass C stocks found there.

## 4.5 Remaining Uncertainties and Future Recommendations

To develop a strong model for sediment C storage in reed beds, it would be preferable to conduct a large-scale sediment
coring campaign across a large number of sites with more sediment samples than we were able to collect in this study.
Developing methods to reliably collect full 1 m soil cores every time and run C analysis on a scale finer than 10 cm
increments would further create very robust data for modeling purposes. As stated in Howard et al. 2014, though most C
storage in vegetated coastal marshes occurs in sediments and belowground biomass, these systems are the least studied.
Retrieving deep soil cores from reed beds and sieving out belowground biomass is cumbersome and time consuming and
largely resource-dependent (e.g. number of people). Allocating more effort on a large-scale sampling campaign would,
nevertheless, generate valuable data that would further contribute to robust models for calculating BC sequestration in
coastal reed beds.

We had trouble retrieving belowground biomass samples from some of the terrestrial and littoral zones. In addition, we did
not retrieve any belowground biomass samples from the littoral zone deeper than 1.4-1.8 m of standing water as this was
logistically infeasible. For future sampling campaigns, we recommend sampling in parts of the littoral zones where standing
water is not so deep or collecting samples from deeper standing water by SCUBA.

The question also remains whether coastal reed bed ecosystems are functioning as true BC ecosystems. BC ecosystems
function as overall C sinks where more C is stored in the ecosystem than lost (Röhr et al. 2016). Further information is
needed showing seasonal variability in reed bed C storage, C isotope presence to tell where the C is coming from, and
greenhouse gas fluxes to get a more comprehensive picture of C cycling in these growing coastal ecosystems. This will then
allow us to better answer the question of whether or not coastal reed beds are acting as net C sinks.

## 5 Conclusions

Understanding C storage in coastal reed ecosystems is crucial for creating more robust estimates of BC. Increased reed
presence in coastal Finland as well as other coastal areas across the world means that the reeds' role in BC storage impacts
management of coastal systems. The results of this study indicate that C storage is occurring in these reed beds, that
environmental context within the different reed bed zones impacts C storage, and that C storage rates are highest in reed bed
sediments. The results further indicate that C storage in coastal reed bed ecosystems is unique from the tidal salt marsh
ecosystems they are typically categorized as in current BC budgets. C storage in these reed beds needs to be considered in

best management practices as authorities look to find new ways to control the reeds' expanding footprint and possibly develop new uses for reed materials.

**Data Availability**

The data used for this project is still in use for other projects within the Center for Coastal Ecosystem and Climate Change Research (www.coastclim.org). Raw data and R-codes can be provided by the corresponding authors upon request.

**Author Contributions**

MW, TJ, AN, and CG planned the campaign; MW performed the measurements; MW analyzed the data; MW wrote the manuscript draft; TJ, AN, and CG reviewed and edited the manuscript.

**Competing Interests**

The authors declare that they have no conflict of interest.

**Acknowledgements**

This research was supported by the Walter and Andree de Nottbeck Foundation and the University of Helsinki Doctoral School and utilized research infrastructure facilities at Tvärminne Zoological Station, University of Helsinki. CG was supported by the Research Council of Finland (project grant 354454). Additional infrastructure facilities at University of Jyväskylä were utilized. This study is part of research from the Center for Coastal Ecosystem and Climate Change Research
(www.coastclim.org). Special thanks to all who helped in the field with data collection: Kurt Spence, Anna Vesanen, Roel Lammerant, Janina Pykäri, and the Onni Talas interns. Special thanks to the following people for the use of R scripts they developed: Sonja Repetti, Quentin Bell, Tom Jilbert, Ana Lindroth Dauner, and Tuomas Junna.

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
