# Peer review of "Carbon Storage in Coastal Reed Ecosystems"

_EGUsphere, 2025_

## Author Comment (AC1)

Reviewer 1 comments:

Review Williamson et al. – Carbon Storage in Coastal Reed Ecosystems

Williamson et al. quantified the carbon (C) storage in both above- and belowground biomass and sediments of common reed beds from different topographical zones along the Pojo Bay, Finland, in the northern Baltic Sea. The results indicate that reed beds, especially in intermittent zones, are important carbon sinks, with higher carbon contents stored in sediment and belowground biomass than aboveground. The manuscript addresses an interesting and relevant topic with significant potential. However, several major issues need to be resolved before it can be considered for publication.

**GREEN: Thank you. We have replied to each comment/suggestion in green. Relevant lines existing in the current text are highlighted in red. We think that the constructive review has significantly improved the paper and thank you for providing detailed feedback.**

General comments

The aim of the study is to investigate C storage in different parts of the reed bed ecosystem across different environmental gradients such as salinity and wave exposure.
RED: Line 12-13: The aim of this spatial study was to quantify how much C is stored in above- and belowground biomass, and sediments in different zones of reed beds along the Pojo Bay system of the northern Baltic Sea in coastal Finland.

**GREEN:** Thank you for bringing this to our attention. The goal of this paper is to quantify C stocks in different reed beds along coastal Finland, similar to what Buczko et al. 2021 did for the German coast. In this study, we did not intend to analyze the impacts of different environmental factors on C stocks, only quantify C stocks found in aboveground biomass (stems and leaves), belowground biomass (roots and rhizomes), and sediments from each of the different reed bed zones across our reed bed sites. Since published data on C stocks in reed beds in our area is lacking, we needed to quantify how much C is in the reed beds and where it is located before anything else. We have changed wording throughout the paper so that this is more clear.

We've changed the text in lines 15, 61, 65, 235, 237, 271-273, 286

We have added a sentence to line 307 (section 4.5 Remaining Uncertainties and Future Recommendations) that reads: "We collected data from sites that covered a range of different environmental gradients such as salinity and wave exposure as a baseline for future investigations that explore how different environmental factors influence C storage in reed beds. At these high-latitudes, coastal reed beds are strongly influenced by seasonal succession, implying that an insightful analysis of environmental drivers requires data collected across different seasons."

- While the results may be described in too much detail, the environmental factors are subsequently mentioned several times but their influence on the carbon storage is not further evaluated. What influence do salinity and wave exposure have on carbon storage? How can the differences between the sites or topographical zones be explained? What is the resulting implication?

  **GREEN:** Please see our response above to the first General Comment.

These points should be more addressed in the discussion.

In addition, I recommend to
- add a schematic overview that highlights or summarizes the differences in carbon storage across the different topographical zones.

  **GREEN:** Thank you, adding a cross section is a great idea. We have created an overall reed bed zone cross section (please see below in our response to your other comment about a cross section). C storage across the different reed bed zones is shown in the corresponding sediment C storage, aboveground biomass C storage, and belowground biomass C storage figures.

- The authors recommend to treat reed beds as distinct ecosystems, and not as conventional salt marsh ecosystems, as they show a great potential for C storage (line 10). However, it is not entirely clear how reed beds differ from salt marshes in terms of carbon stocks. (e.g., line 250-254). It would be good if the differences are highlighted more clearly.
  RED: Lines 47-50 C stocks in reed beds could average around 17.4 kg C m-2 in comparison to estimated global average of C stocks in tidal salt marshes of approximatley 25 kg C m2. Pendleton et al. 2012, Buczko et al 2022, and Silan et al. 2024.

  **GREEN:** We highlight some of the differences between C storage potential in reed beds and tidal salt marsh ecosystems in other parts of the manuscript (for example, lines 47-50), but we can emphasize this point further around lines 50 and 231.

- The definition of "sediment C stock" is unclear (line 161 ff.). Can you please precisely define what the parameter "sediment C stock" refers to? Is it the total amount of C or the total amount of organic matter? In figure 3, you speak of "organic matter". What precisely are the "carbon stocks" given in [g m-2] in figure 4? To which depth interval does the parameter "carbon stock" refer to? You state that the sedimentary carbon stock was integrated over the entire core length (line 105). In general – at least for sediments – the carbon stock or inventory is given for a defined thickness of the surface sediment – e.g. the uppermost 10 cm.

  **GREEN:** Thank you for pointing this out. To clarify, our sediment C stocks refer to TOC (total organic carbon). We ran both TOC and LOI (loss of ignition) on our sediment samples and were

able to calculate a LOI conversion factor so we could calculate TOC for any sediment samples where TOC was not measured. We will discuss this further in the manuscript to clarify. We included figure 3 of the LOI data (Organic Matter %) in addition to the sediment C stock figure as an additional point of interest.

The sediment TOC data was used to calculate the C stock over the entire core length. While some papers look at the uppermost 10cm for sediment C stock data, there is less data published on C stocks from deeper soil depths and these depths can be interesting to look at. Because of this, we wanted to include our full core length data.

We have added sentences to line 129: A LOI conversion factor was calculated from samples where both LOI and TOC were measured. This conversion factor was then used to estimate sediment C stocks for samples where only LOI data was available.

- Furthermore, how is the carbon stock of above- and belowground biomass defined? How comparable are the carbon stocks of the different sample categories?

    **GREEN:** Aboveground biomass is defined as stems and leaves and aboveground biomass is defined as roots and rhizomes. C was measured from subsamples of each of these collected within our 1x1m sampling plots and calculated to g sq m. C stocks.

    These different sample categories are comparable to each other as they are showing a full vertical profile of the C stock within the 1x1m sampling plot (the sediment, the roots/rhizomes within it, and the stems/leaves above it). All C stocks shown are calculated in g m^-2.

    We have added clarification to figure captions for the aboveground biomass and belowground biomass C stocks.

    Line 209: Figure 6: Carbon stocks (g m$^{-2}$) in aboveground biomass (stems and leaves) across all reed bed zones and sites.

    Line 219: Figure 7: Carbon stocks (g m$^{-2}$) in belowground biomass (roots and rhizomes) across all reed bed zones and sites.

- It would be very helpful to include an overview showing a simplified cross-section of a reed bed, including the three different topographical zones and the different sampling sites.

**GREEN:** We think this is a great idea and have worked up the following sketch in Biorender.

Reed Zone Cross Section

[Figure]

Figure X: Reed bed zone cross section. Created in BioRender. Williamson, M. (2025)
https://BioRender.com/99fwu67

Specific comments

Lines 18-30: While the results are described in great detail in the abstract, the overarching implications of the study are missing at this point. The impact of differences in salinity and wave exposure is not explained (see general comment 1).

**GREEN:** Please see our response above to the first General Comment.

Lines 36-38: Carbon (C) and blue carbon (BC) are not introduced in the main text (except for the abstract). Moreover, besides vegetated coastal areas, fine-grained sediments have been shown to represent one of the most important long-term carbon sinks – including e.g. depocenters of mud and tidal flats (e.g., Müller et al., 2025, Biogeosciences, https://doi.org/10.5194/bg-22-2541-2025). These mud areas have received growing attention as important Blue Carbon coastal habitats. Please, therefore mention these as well.

**GREEN:** We have included mud and tidal flats and a citation for Müller et al. 2025 in line 38. Thank you for bringing this paper to our attention. Though mud flats and reed beds are different from one another and we did not conduct any sampling in mud flats, it is nice to have another citation to show the importance of coastal sediments in C storage.

Lines 65-67: What is the basis for hypothesizing that the highest rates of C storage would be found in reed bed sediments?

RED: Lines 248-250: Howard et al. 2014, Fiala 1976. Lines 274-276. Dong et al. 2012

**GREEN:** The basis for this hypothesis is discussed in lines 248-250 and lines 274-276. We have added discussion of these papers earlier in the manuscript so our hypothesis makes more sense.

Line 64: Published findings suggest that most C storage in other types of vegetated coastal systems occur in their sediment and belowground roots and rhizomes (Howard et al. 2014) and that this may hold true for reed beds as well (Dong et al. 2012).

Lines 86-89: This sentence is somewhat difficult to understand. This information about wave exposure (sheltered, semi-sheltered, long and exposed) could be integrated into the overview showing a simplified cross-section of a reed bed (see general comment 4).

**GREEN:** Please see above for the cross section we have drafted. We have also reworded the sentence around lines 86-89 to make it more succinct.

Line 86: Within each reed bed, the littoral zones are along the open water's edge, the intermittent zones are in the middle, and the terrestrial zones are along the edge that pushes into the terrestrial tree line (see Fig X Reed Bed Zone Cross Section).

Lines 90-95: Could this information on sampling be presented visually in the supplementary material? The description does not make it clear how the sampling grid was structured.

**GREEN:** Yes. We have included an example of a map with the 1x1m grid over top of a reed bed. We have included this in the supplementary material as Fig 1

Additional Supplementary Materials

Supplementary Figure 1: Example map of 1x1m grid for random sampling starting point selection

[Figure]

Line 111: What does "when possible" mean in this context?

RED: lines 113-114: soil cores that contained large amounts of clay required soaking in water for 24h back in the laboratory and then additional sieving.

**GREEN:** We explain in lines 113-114 that some soil cores had large amounts of clay in them and required further soaking in water for 24 hours back in the laboratory before additional sieving. We have removed "when possible" in line 111 to reduce confusion as the process is discussed in further detail over the next 2 lines.

Line 120 ff.: Please, state which certified sediment standard reference material was used to assess the quality of the carbon analyses in the sediment samples.

RED: Lines 115-119: samples were run... birch leaf lab standards, USGS88, and USGS91. Lines 127-128: C analysis for sediment samples analzyed at Jyväskylä in the same manner as mentioned above for biomass.

**GREEN:** We mention which standards were used above in lines 115-119 and then clarified that the same process was used for sediment samples in lines 127-128. We have reworded the sentence around 127-128 to clarify.

Lines 127-128: C analysis (total organic carbon, TOC) was also run on 127 of the total 305 sediment samples and analyzed at the University of Jyväskylä using the same standards, equipment, and methods as mentioned above for biomass.

Line 145: The salinity measurement is not included in chapter 2 (Material and Methods).

**GREEN:** We have added a sentence about the method for measuring salinity at each site at the end of line 96.

Line 96: Surface water level salinity was measured at each site with an YSI Pro Solo DO/CT meter

Lines 145-190: The results are described in great detail. It would be better to describe the general trends, differences and/or similarities.

**GREEN:** Thank you for this suggestion. We have attempted to reduce some of the detail listed here and point readers towards the corresponding tables, figures, and supplementary materials for further information.

Line 169 ff.: All three paragraphs of this subchapter start with "We detected statistically significant differences...". This does not read very elegant. Can you rephrase and vary a bit?

**GREEN:** Yes. We have adjusted the wording to be more varied throughout the rest of the text.

Line 210 ff.: You refer to "Carbon stocks (g m$^{-2}$) in belowground biomass" here. Which defined sediment or belowground interval do you refer to here?

**GREEN:** We have added "(roots and rhizomes)" in the caption for Figure 7 to clarify (please see our response in your General Comments section). Roots and rhizomes were sieved from the soil cores and are a separate category of C stock measurement than the sediment C stocks.

Line 239 (GREEN: should be line 230, I believe): You state here that it is imperative "to develop more accurate coastal BC budgets to combat climate change". As mentioned above, please at least also briefly mention the role of fine-grained sediments in the coastal realm in this context as they represent the key and really most important long-term C sinks.

**GREEN:** Yes. Please see our response above for more detailed information about our inclusion of Müller et al. 2025 in line 38. We will avoid discussing mud flat ecosystems in too much depth as this research was conducted in reed beds only but we have included the citation to further show the importance of coastal sediments in C storage.

Lines 234-240: The title of this subchapter is "Dry bulk density and LOI". However, the impact of the environmental factors on the DBD and LOI is not discussed.

**GREEN:** Please see our response above to the first General Comment.

Lines 237 and 246: You mention anthropogenic activities here. Which activities are specifically meant here, and to what extent do they influence DBD and LOI?

**GREEN:** We did not intend to emphasize anthropogenic activities here and we have reworded the section to avoid confusion. We have included some further discussion on the relationship between DBD and OM and talked about how our DBD and LOI results show similar relationships to each other as those found by Cleophas et al. 2024 (higher DBD aligning with lower OM). Our DBDs were highest in terrestrial and littoral zones than intermittent zones which are more buffered. Sediment C stocks and LOI results were highest in intermittent zones.

Line 238: Cleophas et al. 2024 found that higher DBD was associated with lower OM and we see these trends clearly in our own DBD and LOI data.

Lines 238 and 243: The reference USDA-NRCS 2019 is not a scientific publication, but rather a guide for educators. Publications describing the influence of DBD on infiltration and organic matter and nutrient contents should be cited here.

**GREEN:** We have removed the USDA-NRCS 2019 reference at lines 238 and 243-244. We have included citations for Warrence et al. 2003 and Cleophas et al. 2024 in these lines, respectively.

Lines 242-243: To what extent can the differences in sediment carbon stocks between different zones and sites be attributed to differences in DBD? This is not explained here.

**GREEN:** Please see our responses above on the relationship between DBD and OM and our conversion factors between LOI and TOC. We have included a sentence about the trends shown in our LOI, DBD, and sediment C stock figures for further clarification.

Line 245: The C stocks shown in figure 4 correspond with the same LOI and DBD trends found in Clephas et al. 2024 and discussed in the previous section.

Lines 245-246: Here, various environmental factors are mentioned but not discussed further (see general comment 1). How does, for example, different salinity influence sedimentary carbon stocks?

**GREEN:** Please see our response above to the first General Comment. We have adjusted wording throughout the paper accordingly.

Lines 250-254: The calculated sediment carbon stocks are ~8 kg C $m^{-2}$, which is lower than the estimated global average of 25 kg C $m^{-2}$. Does this mean that less carbon is stored in reed bed sediments compared to sediments in tidal salt marshes (see general comment 2)?

**GREEN:** The purpose of these sentences is to compare our results to those from Buczko et al. 2022. Both papers found C stocks in reed bed sediments that are lower than approximate averages known for tidal salt marshes.

Lines 271-273: Again, several environmental factors are mentioned here (anthropogenic activities, wave exposure, changes in soil characteristics), but not discussed with regard to their impact on aboveground biomass carbon stocks. Also, which soil conditions are being referred to here, and what causes these changes?

**GREEN:** Please see our response above to the first General Comment. We have adjusted wording throughout the paper accordingly.

Lines 275-281: This section is a bit misleading and the phrasing should be improved. First, it is stated that removing aboveground biomass reduces the nutrients in a system (line 275). Then, it is argued that removing aboveground biomass can lead to the release of nutrients (line 278). The relationship between removing reed and the release of nutrients should be explained in more detail. Further, how should carbon storage in reed beds be considered in the development of management practices? This is not clearly described here.

**GREEN:** We have reworded this section to avoid confusion.

Lines 274-281+: There has been an increased interest in reed bed management practices focused on removing aboveground biomass (e.g., Finnish-Swedish Interreg BalticReed project co-funded by the European Union). Some of these projects argue that removing aboveground biomass can serve as a way to reducing nutrient inputs to major water bodies, however previous research indicates

aboveground biomass contains relatively low C content (Dong et al. 2012) and our findings support this. The density of aboveground biomass in reed beds can substantially impact erosion and water quality (Horppila et al. 2013) and management practices focused on removing aboveground biomass, such as mowing, can disturb reed bed sediments, releasing nutrients stored in reed beds into the surrounding environment (Güsewell 2003) or even greenhouse gas fluxes (Rietl et al. 2017). In light of this, C storage in reed beds must be taken into account when developing management practices in coastal reed bed ecosystems so the C storage potential of their sediments is not compromised during harvesting processes.

Figure 1: The figure in the bottom left-hand corner (Study Area) is too small to be seen properly. Additionally, the contrast between the two shades of grey is insufficient. One suggestion is to color the water areas blue. Furthermore, the individual stations at the different sites cannot be seen because they overlap.

**GREEN:** We have adjusted the map to better show the location of the reed bed sites.

Updated Fig 1:

[Figure]

Figures 2 and 3: The figures are slightly pixelated, and the symbols are faint. Additionally, the axis labels ('Soil Depth' and 'Dry Bulk Density/Organic Matter') are rather large, and the depth labels ('0-100 cm') are rather squashed.

**GREEN:** We have revised the figures to increase the opaqueness of the symbols, make the axis labels smaller, and increase the spacing between the depth labels.

Fig 2 Updates

[Figure]

Fig 3 Updates

[Figure]

Figures 4 to 7: The figures 4 to 7 show the carbon stocks for the four sample categories. Since the labeling of the y-axis is not the same, it is difficult to compare the values with each other. Moreover, there is something missing in the caption of figure 4: Carbon stocks (g m$^{-2}$) in sediment"s of" all reed bed .....?

**GREEN:** All C stocks are shown in g m^-2 but there is also a figure showing stem count results. We have placed the stem count figure elsewhere so it does not break up the flow of the C stock figures.

Technical corrections

Line 93: "Square" instead of "quadrat"?

**GREEN:** We were not aware that there was some debate about the term "quadrat". This is a term also seen in the publications of other coastal ecologists (example: Hillmann et al. 2020).

Table 1 and 2: The table captions should be placed above the table. In addition, the first lines of the tables should not contain "Table 1" or "Table 2."

**GREEN:** The table captions have been adjusted within the document accordingly

The word "understandable" is used very frequently in the discussion and could be replaced, for example, by „reasonable", "explained by" or "consistent with".

**GREEN:** Wording has been adjusted throughout the paper to provide more variety.

---

## Author Comment (AC2)

The investigation provides information from an understudied ecosystem type. Therefore, the data is valuable. The interpretation of the data is challenging given the limitations in data that could be generated. This calls for a more thorough discussion to avoid having to rely too much on speculation. The method section also requires some clarifications.

GREEN: Thank you for providing feedback. We have replied to each comment/suggestion in green. Relevant lines that already exist in the current text are highlighted in red.

L 65: It would be good to come up with a hypothesis which reed bed zone stores the most carbon. One should be able to derive such a hypothesis from existing literature. Please elaborate.

GREEN: Thank you. We have included additional information to this section based on Reviewer 1's comments as well.

On Line 64 we have added: "Published findings suggest that most C storage in other types of vegetated coastal systems occur in their sediment and belowground roots and rhizomes (Howard et al. 2014) and that this also may hold true for reed beds (Dong et al. 2012). Results from Buczko et al. 2022 suggest that C storage may vary between reed bed zones as well."

L 75 and Table S1: Please add precise geographical coordinates for all sampling locations to Table S1.

GREEN: Thank you. We have now included our coordinates in Table S1, as suggested.

L 105: Good that you were able to sample in this environment. The description sounds as if you succeeded in obtaining undisturbed samples, making possible the calculation of bulk density. Please add a sentence stating this here.

GREEN: Thank you. We have included additional information to this section based on your suggestion.

On Line 105 we have added: "These sampling techniques enabled us to collect undisturbed samples from which bulk density could be calculated."

L 160, Ch. 3.2: In Ch. 2.2 it is stated that these stocks were integrated across the whole core. If these stocks apply to the sampling depth of the cores, which sometimes were sampled down to 50 cm and sometimes to 100cm, how can they be compared. Please state this more clearly, possibly also in Ch. 2.2.

GREEN: We have included additional information on this topic based on Reviewer 1's comments as well. We do agree it should be included earlier in Ch. 2.2. Thank you for the suggestion.

On Line 105 we have added: "Total organic carbon (TOC), LOI, and DBD were measured in 10cm intervals across the length of each sediment core. Sediment C stocks (mass per area) were calculated using LOI, TOC, and DBD for both the whole sampled core length and the upper-most organic matter rich layer. While some papers look at the uppermost 10cm for sediment C stock data, there is less data published on C stocks from deeper soil depths and these depths can be interesting to look at (Yost & Hartemink 2020). Because of this, we wanted to include our full core length data."

L 169 and elsewhere: The term "statistically" is not required. This should be clear.

GREEN: We have removed "statistically" here and reduced its use throughout the rest of the manuscript. Thank you.

Figure 7: Please add "Biomass" before "Belowground C Stocks" at the top of the table.

GREEN: Thank you. We have fixed the title of Figure 7 to read "Belowground Biomass C Stocks"

L 230: Do we really need the expression "to combat climate change"? It is relevant and we want to know. Isn't that good enough?

GREEN: Thank you. We have adjusted the wording to read "as part of climate change mitigation". We have also made the same adjustment to the wording in the abstract on line 34.

L 230: Concerning the expression "unique". Wouldn't "different" be a better expression?

**GREEN:** We have changed the wording from "unique" to "different".

L 231-232: What is that rate? Where is that calculated? Starting which rate does it need to be accounted for? Please avoid unclear statements.

GREEN: Thank you for pointing out this misunderstanding. You are correct, we are not referring to a specific calculated rate of accumulation here.

On line 232, we have adjusted the wording to read: "C storage is occurring in these systems in amounts that need to be accounted for in reed bed management plans."

L 274-281: This paragraph appears out of place here. It provides context that would fit better in the introduction chapter.

Red: Information about reed management programs is already discussed in the introduction section in Lines 51-58. Additionally, the importance of understanding C storage in reed beds as it relates to management practices is stated in the Conclusion section Lines 319-321.

Green: We have made additional adjustments to the paragraph, as requested by both you and Reviewer 1.

On lines 274-277, the paragraph has been reworded to read: "There has been an increased interest in reed bed management practices focused on removing aboveground biomass (e.g., Finnish-Swedish Interreg BalticReed project co-funded by the European Union). Some of these projects argue that removing aboveground biomass can serve as a way of reducing nutrient inputs to major water bodies, however previous research indicates aboveground biomass contains relatively low C content (Dong et al. 2012) and our findings support this."

Discussion: The Discussion picks up some important issues, but given lacking data -and this data is hard to determine- there is some speculation. The discussion would benefit from looking a bit more into processes of C allocation in the sediment. It is probably impossible to come up with C sequestration rates and the question of the stability of the sediment C cannot be answered as well.

GREEN: Thank you for your feedback. You are correct that the goal of this paper is not to determine C sequestration rates or the long-term stability of sediment C in reed beds but to, instead, quantify C stocks in different reed beds along coastal Finland, similar to what Buczko et al. 2021 did for the German coast. Since published data on C stocks in reed beds in our area is lacking, we needed to quantify how much C is in the reed beds and where it is located before anything else. We have changed wording throughout the paper so that this is more clear.

We've changed the text in lines 15, 61, 65, 235, 237, 271-273, 286.

We have added a sentence related to the environmental processes that drive C allocation to line 307 (section 4.5 Remaining Uncertainties and Future Recommendations) that reads: "We collected data from sites that covered a range of different environmental gradients such as salinity and wave exposure as a baseline for future investigations that explore how different environmental factors influence C storage in reed beds. At these high-latitudes, coastal reed beds are strongly influenced by seasonal succession, implying that an insightful analysis of environmental drivers requires data collected across different seasons."

We have made several other changes to our Discussion section based on Review 1's comments. These include:

Line 238: Cleophas et al. 2024 found that higher DBD was associated with lower OM and we see these trends clearly in our own DBD and LOI data.

Line 245: The C stocks shown in figure 4 correspond with the same LOI and DBD trends found in Clephas et al. 2024 and discussed in the previous section.

---

## Author Response (AR2)

Review #3 Comments

Accept with Minor Revisions

The authors have thoroughly addressed the comments and concerns raised in the first review. The manuscript has improved markedly and provides a valuable contribution to the field of carbon storage in vegetated ecosystems, particularly in coastal reed ecosystems. However, I still have some comments regarding the LOI and TOC data. Therefore, I recommend the publication of this publication after some minor revisions.

Thank you for your helpful comments. Please see our responses below in green.

Specific comments

- Lines 113-116: Please add a sentence here, why it is interesting to look also at carbon stocks from deeper soil depths.
    - A sentence on why it is interesting to look at C stocks from deeper soil depths has been added to **Lines 116-119**: "While many earlier studies focus only on the uppermost 10cm for sediment C stock data, Yost & Hartemink (2020) emphasize the importance of generating data from deeper soil horizons as over half of the organic C in the soil column in some settings can be found below 30 cm and the potential for long-term C sequestration increases with depth."

- Lines 137-142: If carbon stocks are defined in terms of TOC, it would be more appropriate to present the measured and calculated TOC data in the main text (Figure 4), while moving the LOI results to the Supplementary Materials. Moreover, loss on ignition is not equivalent to TOC, as mass loss during ignition includes contributions from components other than organic carbon, such as bound water, the combustion of total organic matter (not only carbon), and, depending on temperature, carbonates. LOI should therefore be described as a proxy for organic matter content rather than a direct measure, as it does not represent organic matter quantitatively.
    - We have now clarified LOI's use as a proxy for organic matter content rather than a direct measure and included a new citation on the use of LOI as a method for estimating organic and carbonate content in sediments (Heiri et al. 2001). Most samples had both LOI and TOC measurements but LOI was used as a proxy for organic matter content when there were not TOC measurements as well. I have added a visualization of the regression that was used to generate the conversion factor of 1/2.1 to the supplement

(Figure S2) to show the strong relationship between these parameters and kept Fig. 4 as our LOI data.

- Wording has been added to **lines 144-146**: "For samples with no direct TOC measurement, a LOI% to TOC% conversion factor of 1/2.1 was applied, as calculated in R from the slope of the linear regression line for all samples with both LOI and TOC data (Figure S2)."
- Citation has been added to **line 136** and the **References** (Heiri et al. 2001)
- A visualization of the regression used to generate the conversion factor of 1/2.1 has been added to supplementary materials (**Figure S2**). Figures pre-exisiting in the Supplementary materials section have been re-numbered accordingly and their references updated throughout the manuscript (**Lines 146, 190, 301**).

- Lines 139-140: Please add how exactly you have calculated the sediment carbon stocks.
    - The sediment C stocks were calculated in R using several different formulas. I agree that including more details about how we calculated the sediment C stocks is useful.
        - I have tried to summarize this efficiently and **lines 142-144** now read: "Sediment C stocks were calculated for a given sediment thickness (either *0 cm to 10 cm*, or *0 cm to bottom of core*) by multiplying the TOC content (%) by the DBD ($g\ cm^{-3}$) and the sample volume ($cm^3$) in a hypothetical sediment column of surface area 1 $cm^2$. Data were then converted to $g\ C\ m^{-2}$ for the given thickness of the layer (Table S4)."
        - Sediment increment has been added to **Figure 5, Table 1, and Table 2**: (0 cm to bottom of core)
        - 0 cm to 10 cm sediment stock columns have been added to **supplemental table S4**.

- Lines 141-142: Please also add the formula for the LOI conversion factor.
    - **Line 144-146** now says: "For samples with no direct TOC measurement, a LOI% to TOC% conversion factor of 1/2.1 was applied, as calculated in R from the slope of the linear regression line for all samples with both LOI and TOC data (Figure S2).

- Line 209 and 219: Please add the definition of aboveground and belowground biomass not only in the figure captions, but also in the main text.

- - o I have now added aboveground and belowground biomass definitions in the main text on the following lines: **line 28, line 29, and line 65-66**.

- Figure 2: This may be a matter of personal preference, but the use of color in this figure could enhance readability.
  - o I have replaced **Figure 2** with a color version. Please let me know if you prefer this one to the black and white version.

Technical corrections

- Lines 38-39: The terms carbon (C) and blue carbon (BC) still had not be defined in the main text. The same is true for loss on ignition (LOI) in line 112.
  - o Thank you for catching this. The terms are now defined on the following lines: C is now defined on **line 39**, BC is now defined on **line 40**, and LOI is now defined on **line 114**.

- Line 40: Please replace "Muller" by "Müller".
  - o Thank you. "Muller" has now been replaced with "Müller" on **line 41**.

- Lines 278-280: Please replace "correspond with" by "correspond to".
  - o Thank you. **Line 283** now says "correspond to".

- Lines 283-284: Please modify as follows: Our sediment C stocks averaged between approximately 6 and 16 C kg m$^{-2}$ across the different sites, compared to the values of 17.4 C kg m$^{-2}$ reported by Buczko et al. (2022).
  - o Sounds good. **Lines 287-289** now read: "Our sediment C stocks averaged between approximately 6 and 16 C kg m$^{-2}$ across the different sites, compared to the values of 17.4 C kg m$^{-2}$ reported by Buczko et al. (2022)."

- Lines 288-295: The parentheses around the publication year are missing in the references to Buczko et al. (2022).
  - o Thank you. Parentheses have been added around the publication year for Buczko et al. (2022) on **lines 289, 292, 296, and 297**.

- Line 296: Since the previous Figure 5 is now in the Supplementary Material, the reference here is incorrect.

- Thank you. The reference has been corrected to read Figure S3. **Line 301** now says "(Figure S3 and Tables 1 and 2)"

---

## Author Response (AR3)

Editor Comments and Response

- Line 219 / Figure 2 caption: Please remove the weblink and replace it with appropriate copyright information, as weblinks are not considered permanent sources of information. Thank you. We have removed the URL and cited the figure as suggested by BioRender protocol. **Line 219/Figure 2 Caption** now reads: Figure 2: Reed bed zone cross section. Created in BioRender. Williamson, M. (2025)

- Table 1: There appears to be no strong reason to split this table into two parts. With a small redesign (e.g., using a smaller font), the table could likely be merged into a single version that fits on one page.
We have adjusted font size on **Table 1** and it now fits onto one page.

- Data availability statement: The current wording (e.g., "The data used for this project is still in use...") is not suitable. With publication, the data automatically becomes publicly available. Please revise this section to clearly describe how the data can be accessed, for example by providing a DOI to a public repository. Alternatively, the data may be included as supplementary Excel files.

We have uploaded data associated with this manuscript to a public repository and listed the DOI in our revised Data Availability section. **Line 360/Data Availability** section now reads: The data used for this project can be accessed via https://doi.org/10.5281/zenodo.18537690 (Williamson, 2026)